# Mechanism of connexin channel inhibition by mefloquine and 2-aminoethoxydiphenyl borate

Pia Lavriha[1,2☉], Yufei Han[1,2☉], Xinyue Ding[1,2], Dina Schuster[1,3], Chao Qi[1,2,4], Anand Vaithia[1], Paola Picotti[3], Volodymyr M. Korkhov[1,2]*

1 Laboratory of Biomolecular Research, Paul Scherrer Institute, Villigen, Switzerland, 2 Institute of Molecular Biology and Biophysics, ETH Zurich, Zürich, Switzerland, 3 Institute of Molecular Systems Biology, ETH Zurich, Zürich, Switzerland, 4 Medical Research Council Laboratory of Molecular Biology, Cambridge, United Kingdom

☉ These authors contributed equally to this work.
* volodymyr.korkhov@psi.ch

**Data Availability Statement:** The cryo-EM density maps and model coordinates have been deposited to the Electron Microscopy Data Bank and Protein

## Abstract

Gap junction intercellular communication (GJIC) between two adjacent cells involves direct exchange of cytosolic ions and small molecules via connexin gap junction channels (GJCs). Connexin GJCs have emerged as drug targets, with small molecule connexin inhibitors considered a viable therapeutic strategy in several diseases. The molecular mechanisms of GJC inhibition by known small molecule connexin inhibitors remain unknown, preventing the development of more potent and connexin-specific therapeutics. Here we show that two GJC inhibitors, mefloquine (MFQ) and 2-aminoethoxydiphenyl borate (2APB) bind to Cx32 and block dye permeation across Cx32 hemichannels (HCs) and GJCs. Cryo-EM analysis shows that 2APB binds to "site A", close to the N-terminal gating helix of Cx32 GJC, restricting the entrance to the channel pore. In contrast, MFQ binds to a distinct "site M", deeply buried within the pore. MFQ binding to this site modifies the electrostatic properties of Cx32 pore. Mutagenesis of V37, a key residue located in the site M, renders Cx32 HCs and GJCs insensitive to MFQ-mediated inhibition. Moreover, our cryo-EM analysis, mutagenesis and activity assays show that MFQ targets the M site in Cx43 GJC similarly to Cx32. Taken together, our results point to a conserved inhibitor binding site in connexin channels, opening a new route for development of specific drugs targeting connexins.

## Introduction

Connexins are transmembrane proteins which assemble into hexameric hemichannels (HCs) at the cell surface allowing the exchange of small solutes (ions and molecules of molecular weight below ~1.5 kDa) between the cell cytoplasm and the extracellular environment [1]. Two HCs at the surface of the neighboring cells can dock in a head-to-head fashion, forming gap junction channels (GJCs) [2]. The GJCs connect the cytoplasms of neighboring cells and ensure their electrochemical and metabolic coupling in a process called gap junction intercellular communication (GJIC) [1–3]. There are 21 connexin isoforms in humans, with distinct

Data Bank with the following accession numbers:
PDB ID 8QJF, EMD-18446; PDB ID 8QK6, EMD-
18457; PDB-ID 8QJH, EMD-ID 18447; PDB-ID
8QKI, EMD-18463; PDB-ID 8QKO, EMD-18468.
The mass spectrometry proteomics data have been
deposited to the ProteomeXchange Consortium via
the PRIDE partner repository with the dataset
identifier PXD056242. All data required to evaluate
the conclusions in this paper are present in the
paper and in the Supplementary Materials.

**Funding:** The work was supported by the Swiss
National Science Foundation (grant 184915) to V.
M.K. the funders had no role in study design, data
collection and analysis, decision to publish, or
preparation of the manuscript.

**Competing interests:** The authors have declared
that no competing interests exist.

tissue distribution, channel properties and regulatory mechanisms [2]. These channels are responsible for a variety of essential functions including cardiac contraction [4], central and peripheral nervous system signaling [5,6], tissue differentiation [7], immunity [8] and cell growth [9]. Connexin-mediated solute permeation must be tightly controlled, as highlighted by the numerous pathologies stemming from connexin mutations or dysregulation [10]. The diseases linked to connexin channels, or connexinopathies, include oculodentodigital dysplasia [11], palmoplantar keratoderma [12], keratitis-ichtyosis-deafness (KID) syndrome [13], multiple types of cancer [9,14] (reviewed in [15]).

Given the crucial role connexin channels play in health and disease they have emerged as candidate targets for drug development [16]. Inhibition of connexin GJCs and HCs, such as Cx26 and Cx43, has been suggested to be a promising strategy in treatment of KID syndrome (Cx26) [17], arrhythmias (Cx43) [18], and in acceleration of wound healing (Cx43) [19]. To this end, connexin channels have been shown to be inhibited by RNA, peptides, antibodies, and small molecules. The latter are particularly attractive as classical pharmacological agents, and a substantial amount of work has been done to characterize small molecule drug-mediated connexin inhibition. For example, glycyrrhetinic acid and its derivatives [20], long-chain alcohols [21,22], fatty acids [23], quinine and its analogs [24–26] have been identified as inhibitors of connexin HC and GJC function. The substantial problem with all of these inhibitors lies in their limited connexin isoform selectivity [27]. Furthermore, despite the availability of high-resolution structures of several connexin channels, the structural basis of connexin channel inhibition by small molecule inhibitors (as well as any other types of inhibitors) is lacking.

Among the known small molecule connexin inhibitors are mefloquine (MFQ) and 2-aminoethoxydiphenyl borate (2APB) (**Fig 1A**). MFQ, a compound on the World Health Organization (WHO) Model List of Essential Medicines, is an antimalarial compound that targets the *Plasmodium falciparum* ACBP [28]. Administration of MFQ is associated with neurological [29], neuropsychiatric [30], hepatic [31], and gastrointestinal side effects [28]. MFQ has been shown to be a specific inhibitor of Cx36 and Cx50, capable of blocking Cx26, Cx32, and Cx43 at higher concentrations [24]. In combination with donepezil or amitriptyline, MFQ is currently tested in pre-clinical models for treatment of connexin-related disorders linked to Alzheimer disease (Cx36) and neuropathic pain (Cx43), respectively [16,32–34]. The second connexin inhibitor, 2APB, has been shown to have cardioprotective, anti-inflammatory, and anti-oxidative properties [35,36], inhibiting a range of connexin channels (Cx26, Cx36, Cx32, Cx40 and Cx45 HCs and GJCs) among other targets [37,38]. Understanding the molecular basis of connexin inhibition by MFQ and 2APB would enable structure-based discovery and development of drugs that target these channels in a variety of connexin-related pathologies. Moreover, development of highly specific connexin-targeting drugs with fewer side effects would be highly desirable.

To gain insights into the mechanism of connexin channel inhibition by small molecules, we analysed the structures of Cx32 in the presence of MFQ and 2APB and performed functional analysis of the drug effects *in vitro* and *in cellulo*. Our results reveal two previously undescribed modes of Cx32 channel inhibition by these two drugs. The primary mechanism of MFQ-mediated inhibition involves a conserved drug binding site deep within the pore of Cx32, Cx43 (this study) and Cx36 (a companion study) [39]. Furthermore, 2APB additionally engages a distinct binding site close to the pore entrance of Cx32.

## Results

### Cx32 inhibition by MFQ and 2APB

The effects of MFQ and 2APB on Cx32 have been previously investigated. MFQ was shown to inhibit Cx32 GJC-mediated coupling in RIN cells [40], and reduce the junctional currents in

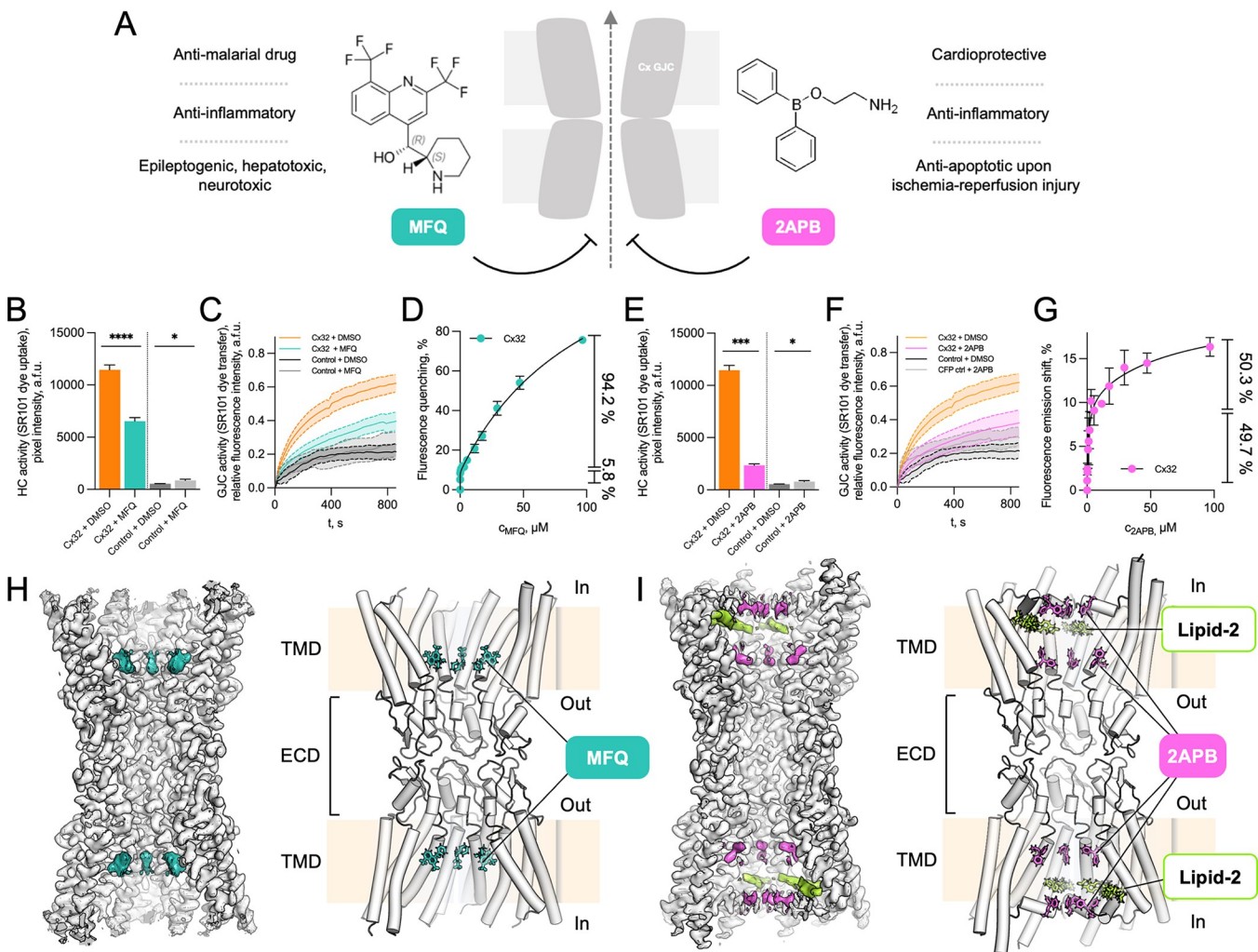

**Fig 1. Analysis of Cx32 channel inhibition by MFQ and 2APB.** (**A**) Schematic representation of connexin inhibition by MFQ and 2APB. (**B**) Dye uptake of Cx32 HC and mock-transfected control cells upon treatment with MFQ or DMSO (drug solvent) (Cx32: MFQ, n = 1183; DMSO, n = 1109; Control: MFQ, n = 1047; DMSO, n = 1016). The HC dye-uptake values were compared using Games-Howell's multiple comparisons test: ****—P < 0.0001, ***—P < 0.001, **—P < 0.01, *—P < 0.05, ns–P > 0.05. *n–number of measured cells; all experiments were performed in experimental triplicates.* (**C**) GJC permeability in Cx32 and mock-transfected control cells upon MFQ, or DMSO treatment (Cx32: DMSO, n = 12; MFQ, n = 16; Control: DMSO, n = 7; MFQ, n = 13). (**D**) Cx32 tryptophan fluorescence quenching upon MFQ treatment (n = 3). (**E**) Dye uptake of Cx32 HC and mock-transfected control cells upon treatment with 2APB or DMSO (Cx32: 2APB, n = 911; DMSO, n = 1109; Control: 2APB, n = 1047; DMSO, n = 1016). *n–number of measured cells; all experiments were performed in experimental triplicates.* (**F**) GJC permeability for Cx32 and mock-transfected control upon 2APB, or DMSO treatment (Cx32: DMSO, n = 12; 2APB, n = 15; Control: DMSO, n = 7; 2APB, n = 12). (**G**) Spectral shift of tryptophan fluorescence upon 2APB titration (n = 3). The values (%) on the right side in **D** and **G** indicate the distribution of the calculated low and high affinity binding sites. All data is represented as mean ± SEM. (**H**) Side views of cryo-EM map and model of Cx32 GJC in complex with MFQ. (**I**) Same as **H**, for Cx32-2APB (level = 4σ). Additional densities in **H** and **I**, interpreted to be MFQ and 2APB respectively, are illustrated with colored features. TMD–transmembrane domain; ECD–extracellular domain; In–intracellular side; Out–extracellular side.

N2A cells [24,40]. 2APB was shown to inhibit purified Cx32 HCs and Cx32 GJC-mediated dye-coupling in HeLa cells [37], but did not affect junctional conductance in Cx32-coupled N2A cells [38]. To assess the effect of these drugs on Cx32 HCs and GJCs we performed HC dye uptake assays and gap-fluorescence recovery after photobleaching (gap-FRAP) experiments (**Fig 1B, 1C, 1E and 1F**). We used HEK293F cells which express Cx32 at the plasma membrane [41] and form functional gap junctions (S1 Fig in S1 File). Both drugs inhibited the Cx32 HC activity (**Fig 1B and 1E**) and the GJC-mediated dye transfer (**Fig 1C and 1F**, S4 Fig A-B in S1 File).

HEK293 cells are known to express pannexin-1 (Panx1) [42], which could potentially influence the observed dye uptake. To determine whether Panx1 expression is modified by over expression of Cx32, we performed mass spectrometry-based cell surface biotinylation assays. The results confirmed that although Panx1 is indeed expressed in our cells, transfection with the Cx32 expression plasmid does not increase Panx1 expression. The relative expression levels of Cx32 and Panx1 in the control cells were (in arbitrary units): undetected and 41429.81 ± 20171.99, respectively. In Cx32- transfected cells, Cx32 and Panx1 levels were 674598.69 ± 58360.1 and 30106.35 ± 9649.86, respectively (S2 File). Thus, we interpret the observed signals in the HC assays as a consequence of Cx32 expression at the plasma membrane.

Both drugs, MFQ and 2APB, were able to bind to the purified Cx32 (S5 Fig in S1 File), as shown by the intrinsic tryptophan fluorescence quenching-based binding assays (**Fig 1D and 1G**, S6 Fig in S1 File). Analysis of the MFQ binding data using non-linear regression with a one-site specific binding model resulted in an estimate of a binding constant of 43.1 μM (**Fig 1D**). For 2APB, fitting the tryptophan emission shift data to a two-site specific binding model resulted in apparent high and low affinity binding constants of 1.05 μM and 75.1 μM, with both sites occupied approximately equally (**Fig 1G**).

## Structural analysis of Cx32 GJCs in complex with MFQ or 2APB

To determine the location of the drug binding sites in Cx32 and to assess the effects of the drugs on the channel structure, we prepared single particle cryo-EM samples of purified Cx32 (S5 Fig in S1 File) in the presence of MFQ and 2APB. Using cryo-EM analysis and image processing we obtained 3D reconstructions of the Cx32 GJCs in D6 symmetry bound to MFQ at 2.91 Å resolution (S7 and S9A Fig in S1 File) and to 2APB at 2.86 Å resolution (S8, S9C, S12A and S12C Fig in S1 File, S1 Table in S1 File). The quality of the density maps was sufficient to resolve all of the features of the Cx32 GJC structure [41], namely the transmembrane helices (TM1-4), extracellular loops 1 and 2 (ECL1-2) and lipid or detergent-like densities on the protein-lipid interface (S10A and S11A Fig in S1 File). In addition to these expected structural elements each of the maps revealed features specific to the added drugs, as detailed below.

Focusing on the Cx32 HC particles (S7B and S8B Fig in S1 File) allowed us to obtain the cryo-EM reconstructions of Cx32 HCs in the presence of MFQ and 2APB at 3.46 Å and 3.14 Å resolution, respectively, in C6 symmetry (S7C, S8C, S9B, S9D, S10B and S11B Fig in S1 File). Since the analysis of Cx32 HC in complex with 2APB and MFQ revealed differences to Cx32-apo HC [41] that were difficult to unambiguously interpret, we have focused our analysis on the GJC structures (S13 Fig in S1 File).

## Additional densities corresponding to MFQ and 2APB in Cx32 reconstructions

Addition of MFQ to Cx32 GJC did not alter the overall conformation of the channel, with an RMSD of 0.447 Å between the Cx32-apo [41] and the Cx32-MFQ (**Fig 2**). The N-terminal helix (NTH) remained unresolved, similar to the Cx32-apo GJC structure (**Figs 1H and 2B**). However, an additional density element was clearly present in the pore region corresponding to the previously described lipid-1 site. We interpret this density as the site of MFQ binding, and thus this region of Cx32 is referred to as "site M".

Although overall the difference between Cx32-apo and Cx32-2APB was very small in the transmembrane and extracellular domains (0.344 Å), addition of 2APB caused a clear conformational change in the Cx32 GJC (**Figs 1I and 2C**). The NTH, flexible in the apo- and MFQ-bound forms of the channel, changed to a conformation closely resembling the NTH

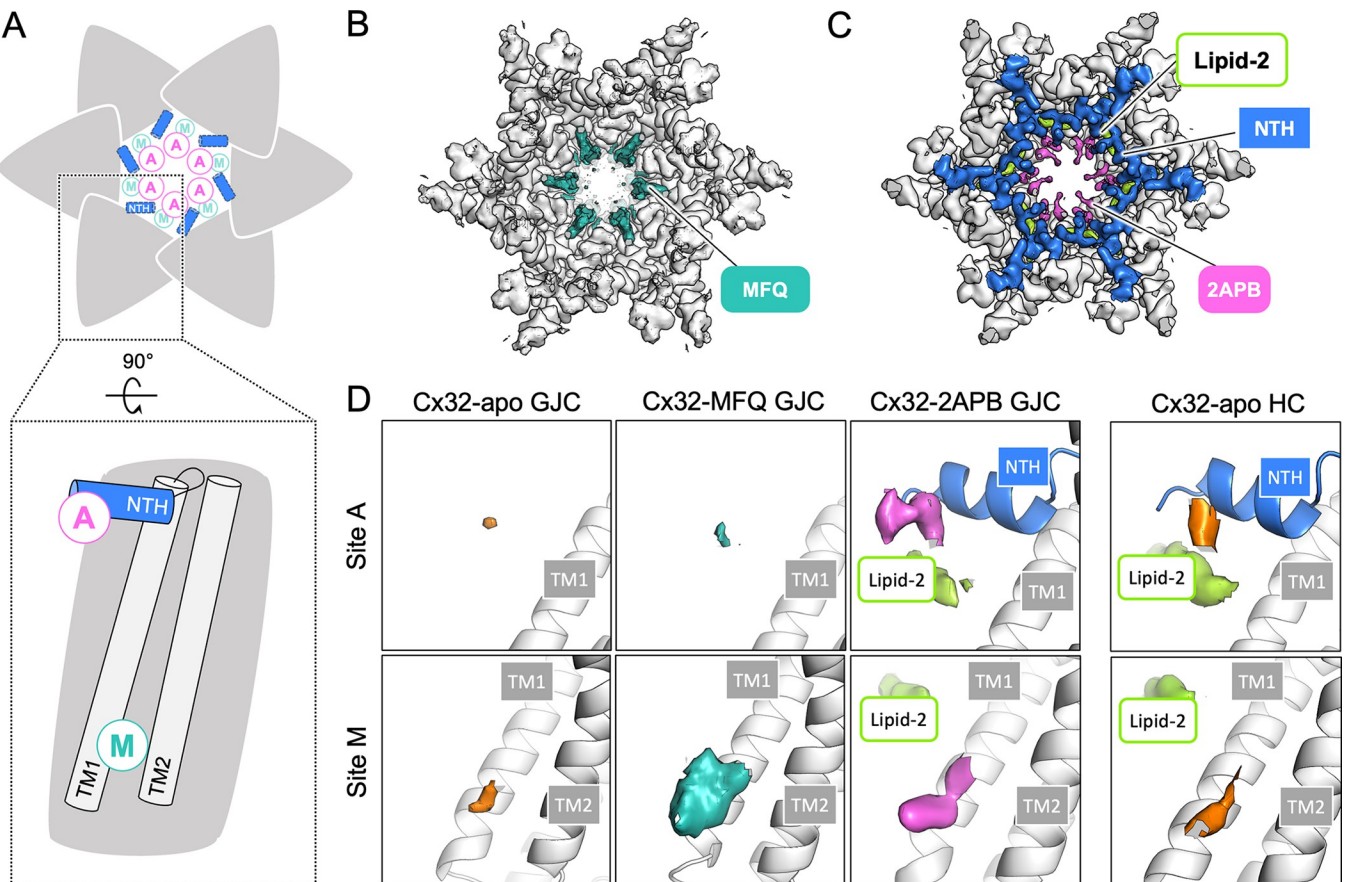

**Fig 2. Locations of the MFQ and 2APB binding sites in the Cx32 GJC.** (**A**) Schematic representation of putative MFQ and 2APB binding sites A and M in the Cx32 GJC. (**B-C**) Views of the Cx32-MFQ (**B**) and Cx32-2APB maps (**C**), contoured at 4σ. (**D-E**) A comparison of the densities at sites A and M in Cx32-apo GJC and Cx32-MFQ GJC, Cx32-2APB GJC (**D**) and Cx32-apo HC (**E**) (all maps contoured at 3σ). TM–transmembrane helix, NTH–N-terminal helix.

conformation in the Cx32-apo HC structure [41]. An additional density between the NTH regions of the neighboring Cx32 monomers likely corresponds to 2APB. We refer to this region of the protein as "site A". Moreover, a poorly resolved additional density is present in the site M (**Fig 2D,** S12 Fig in S1 File), potentially indicative of weak binding of 2APB to this drug binding pocket, in addition to site A.

To investigate the MFQ and 2APB binding sites, we have compared the site A and site M densities in the reconstructions of Cx32-apo, Cx32-MFQ and Cx32-2APB GJCs (**Fig 2D,** S12 Fig in S1 File). Upon addition of MFQ, the only change in Cx32 GJC density occurs at site M. This density is large enough to accommodate the quinoline part of the molecule. However, there is no clearly defined density which fits the piperazine moiety of either (+)-(11*R*, 12*S*) or (-)-(11*S*, 12*R*) enantiomer of MFQ, both of which were present in the sample (S12A Fig in S1 File). As accurate positioning of MFQ is not possible, we modelled the MFQ based on its conformation in a complex with Cx36 GJC, for which a high-resolution 3D reconstruction is available. In the case of 2APB, the site M density only allows approximate placement of the drug, whereas site A is sufficiently resolved for more reliable modeling of the molecule (S12B Fig in S1 File).

To assess the variability in drug binding among individual Cx32 subunits we performed protomer-focused classification (PFC) [43] of Cx32 GJC in complex with MFQ and 2APB (S14

Fig in S1 File). The classification did not result in 3D classes with improved MFQ density and did not allow the separation between the two enantiomers. However, classes 1 and 5 revealed that not all subunits were occupied by the drug, whereby class 1 also contained a somewhat better resolved N-terminus lining the Cx32 pore. Similar to MFQ, PFC did not yield a class with either of the 2APB binding sites better resolved. The low resolution of the 3D reconstructions prevented us from quantitatively assessing the differences in site A and site M occupancies. Nevertheless, the classification confirmed that there is some variability in the NTH of the subunits constituting the drug-bound GJC.

## Binding of 2APB to site A

The NTH in Cx43 has been previously shown to bind lipids, presumably stabilizing the NTH in the closed conformation [43,44]. The site A in Cx32-2APB is located at this region, between the NTH of the neighboring subunits (**Fig 3A,** S15A Fig in S1 File). The phenyl rings of 2APB are in contact with the hydrophobic side of the amphipathic NTH, in close proximity to the amino acids M1, G5, and L9. The lipid-2 density observed previously in the Cx32-apo structure [41] is located nearby and may contribute to 2APB binding. As we have imposed the D6 symmetry, we have modelled six 2APB molecules occupying the site A. Considering the apparent affinity of 2APB for Cx32 and the drug concentration used for cryo-EM structure determination, modeling of six 2APB molecules is appropriate (keeping in mind the results of the PFC analysis, which suggest some variability in site A occupancy).

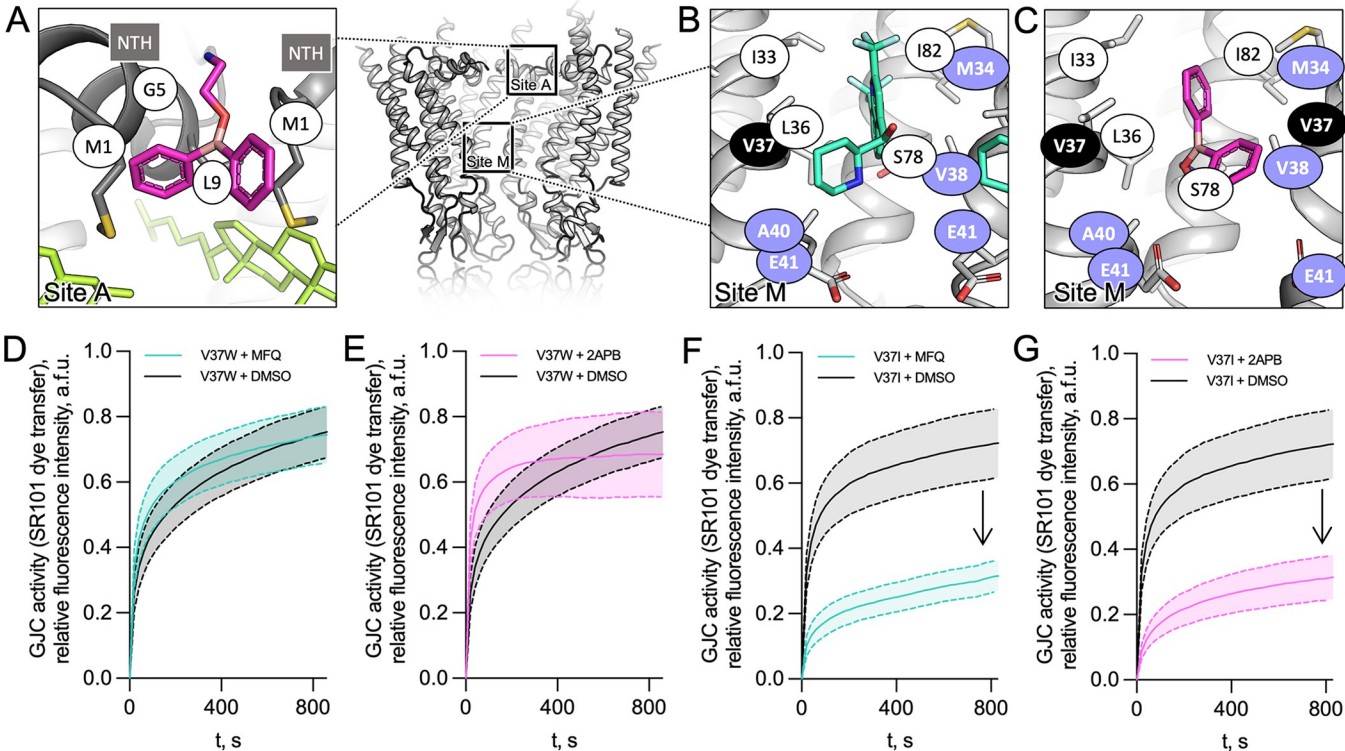

**Fig 3. Molecular details of the inhibitor binding sites in Cx32.** (**A**) Locations of site A and M are shown with the boxes (*right*). Left: site A. Residues (side chains shown as sticks) constituting the site A, within 4 Å of the bound 2APB molecule, are shown with white labels. The sterol molecule (green) is located in close vicinity of the site A. (**B-C**) Site M. Side chains of the residues in close proximity of the MFQ (**B**) or 2APB (**C**) are shown as sticks. Purple labels indicate residues linked to CMT1X disease. (**D-G**) GJC permeability of V37W and V37I Cx32 mutants upon treatment with MFQ (**D**, **F**) and 2APB (**E**, **G**). The GJC activity values are shown as mean ± SEM; for **D**, n = 20 (DMSO) and n = 18 (MFQ); for **E**, n = 20 (DMSO) and n = 16 (2APB); for **F**, n = 17 (DMSO) and n = 19 (MFQ); for **G**, n = 17 (DMSO) and n = 20 (2APB).

## Binding of MFQ and 2APB to site M

The site M is formed by I33, L36, V37, A40, E41, S79, and I82 of one subunit, and M34 and V38 of the neighbouring subunit, and is thus largely hydrophobic (**Fig 3B and 3C**, S15B and S15C Fig in S1 File). However, S79 and E41 may be involved in hydrogen bond formation with the hydroxyl and/or amine group of MFQ, and the amine group of 2APB. Our data clearly implicate this site in functional regulation of Cx32. Consistent with this, mutations of several amino acid residues located within this site (M34, V38, A40, and E41) are involved in the onset of the X-linked Charcot-Marie-Tooth (CMT1X), a Cx32-linked connexinopathy [5], highlighting the functional importance of site M in channel function under physiological and pathophysiological conditions.

To biochemically test whether inhibition of Cx32 by MFQ and 2APB is mediated by site M binding we generated single amino acid residue mutants V37T, V37I, V37W, V38W, S78F, E41D, and E41A. The mutants were aimed to disrupt the binding site by either steric hinderance, change of electrostatic properties, and/or disruption of polar bonds between the drug and the binding site. Each of the mutants could be successfully expressed in HEK293F cells (S1C Fig in S1 File), albeit at different expression levels. Except for V37T, all mutants formed the GJ plaques to approximately an equal extent (S1A-B Fig in S1 File). HC dye uptake assays showed that all of the mutants have a decreased basal permeability to the dye compared to wild-type Cx32, likely reflecting the suboptimal solute translocation pathway properties in the mutant pores (S4C-S4K Fig in S1 File). The effect of the two drugs on the mutants is reduced (with the exception of V37T and V37I and MFQ), suggesting that mutating these amino acid residues disrupts drug binding (S4 Fig in S1 File). V37T and V37I are less bulky and likely do not alter the site M properties sufficiently to prevent drug binding, particularly when other amino acid residues in this site are involved in the drug interaction.

To probe the effects of MFQ and 2APB on GJC function, we selected V37I and V37W mutants, based on their distinct sensitivity to the two drugs in HC assays. Consistent with the HC inhibition of the HC result, the gap-FRAP experiments showed that V37W GJCs were not sensitive to MFQ and 2APB treatment, pointing to the disruption of site M as a drug binding site (**Fig 3D and 3E**, S4A and S4B Fig in S1 File). In contrast, drug sensitivity of V37I in GJC permeability assays was similar to that of the wild-type Cx32 (**Fig 3F and 3G**, S4A and S4B Fig in S1 File). Thus, substitution of V37 in site M with a bulky side chain disrupts MFQ and 2APB sensitivity in cellular assays of Cx32 HC and GJC function, indicating that site M is the primary inhibitory site for small molecule inhibitors. Site A additionally is engaged by 2APB, causing a conformational change and stabilizing the NTH.

## Influence of drug binding on Cx32 GJC pore characteristics

MFQ and 2APB have distinct effects on Cx32 channel conformation. Binding of 2APB to site A and the concomitant rearrangement of the NTH reduces the pore radius from ~15 Å to only ~2 Å (**Fig 4A and 4B**). Interestingly, binding of 2APB to site M does not result in a decrease of the pore radius: a substantial gap remains between the drugs arranged radially within the pore. Instead of a mere steric obstruction, 2APB binding changes the electrostatic properties of the channel pore making it more hydrophobic (**Fig 4C and 4E**, S15D Fig in S1 File). Likewise, MFQ binding to site M decreases the pore diameter only slightly, concomitant with the changes in the electrostatic potential of the pore (**Fig 3D**, S15D Fig in S1 File). Binding of two "rings" of MFQ (or 2APB) to the site M regions of connexons linked within the GJC introduces two hydrophobic barriers in the way of the solutes moving through the channel. Thus, although the pore dimensions appear to be unchanged due to only a minor steric hindrance, free movement of ions or small molecules through the newly formed hydrophobic drug ring

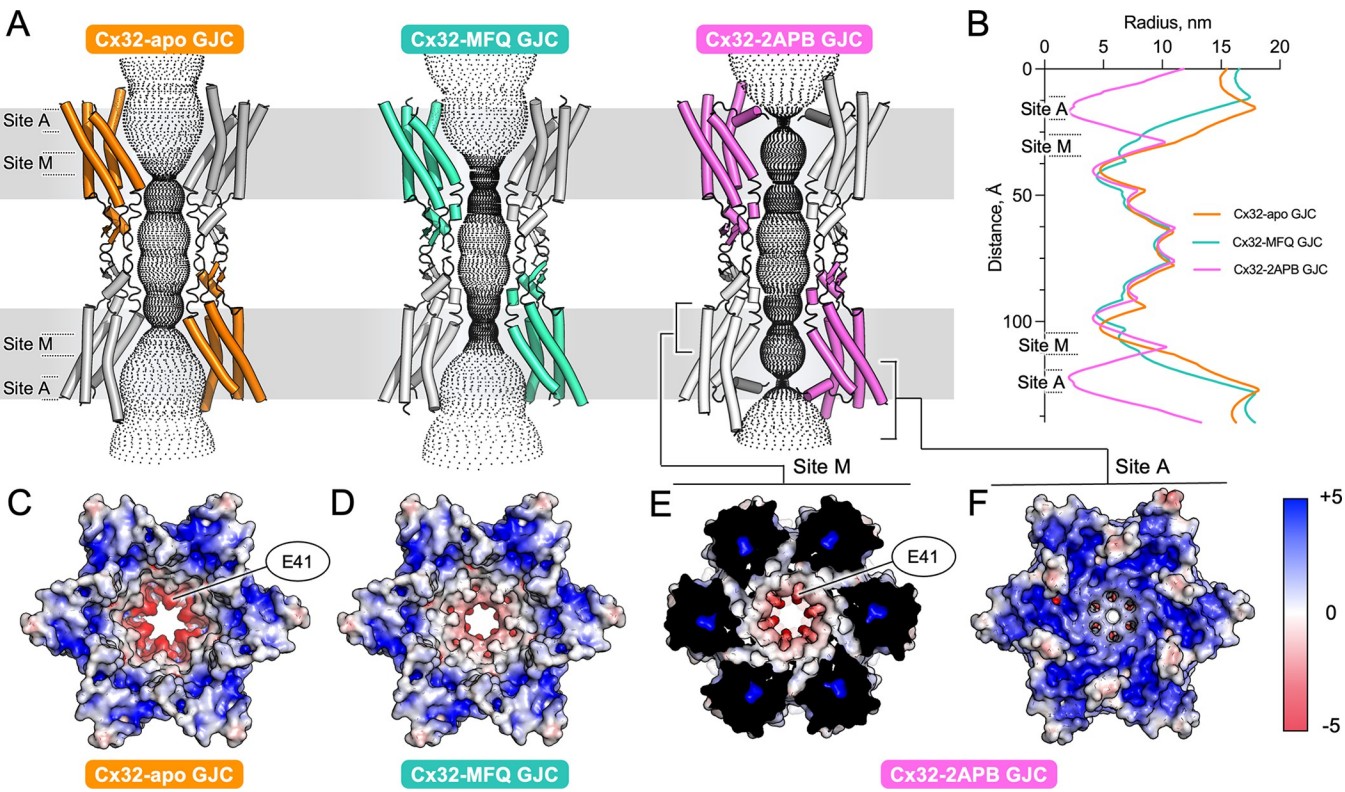

**Fig 4. Effect of drug binding to site A and M on Cx32 GJC pore properties.** (**A**) Diffusive pathways for the solutes of Cx32 and Cx32 in complex with MFQ or 2APB, calculated using HOLE. (**B**) Pore radii along the pore, calculated using HOLE (shown in **A**). (**C-D**) Electrostatic surface potential representations of Cx32-apo GJC (**C**), Cx32-MFQ (**D**). (**E**) The clipped view of Cx32-2APB revealing the site M. (**F**) The unclipped view of Cx32-2APB (as in **C** and **D**).

inside the pore is restricted. Moreover, binding of the drugs to site M likely neutralizes the residue E41, a polar residue within the connexin pore, playing a key role in voltage gating of the channel [45]. Thus, the combined steric and electrostatic effects of drug-binding to site M likely underlies the small molecule-mediated inhibition of Cx32.

## MFQ binds to site M of Cx43 GJC

Our functional and structural data on MFQ and 2APB suggest that site M likely serves as the primary site for inhibitor action in Cx32 GJC. To determine whether site M is a universal inhibitory drug binding pocket in connexins, we analyzed the effects of MFQ on Cx43 using cryo-EM (**Fig 5A–5C**) and functional analysis in cell-based activity assays (**Fig 5D**).

MFQ has been previously shown to inhibit GJC activity in Cx43-coupled N2A [24] and RIN cells [40]. Using cellular assays in HEK293F cells we observed its inhibitory effect on Cx43 GJC-mediated dye transfer (**Fig 5D**). Cryo-EM structure determination of the Cx43 GJC in complex with MFQ at 3.7 Å resolution revealed a state of Cx43 similar to that observed previously in Cx43-apo [43,44] (**Fig 5A**, S16-S18 Fig in S1 File). However, similar to Cx32 GJC, a prominent additional density corresponding to MFQ was observed within the region corresponding to site M (**Fig 5B**). The limited quality of site M density only allowed us to model MFQ using the higher resolution structure of Cx36-MFQ as a template (**Fig 5A**; PDB ID: 8QOJ). Similar to Cx32, while leaving a substantial opening inside the pore, MFQ binding causes radical changes in the electrostatic properties of the solute translocation pathway, generating two hydrophobic barriers within the channel (**Fig 5E**).

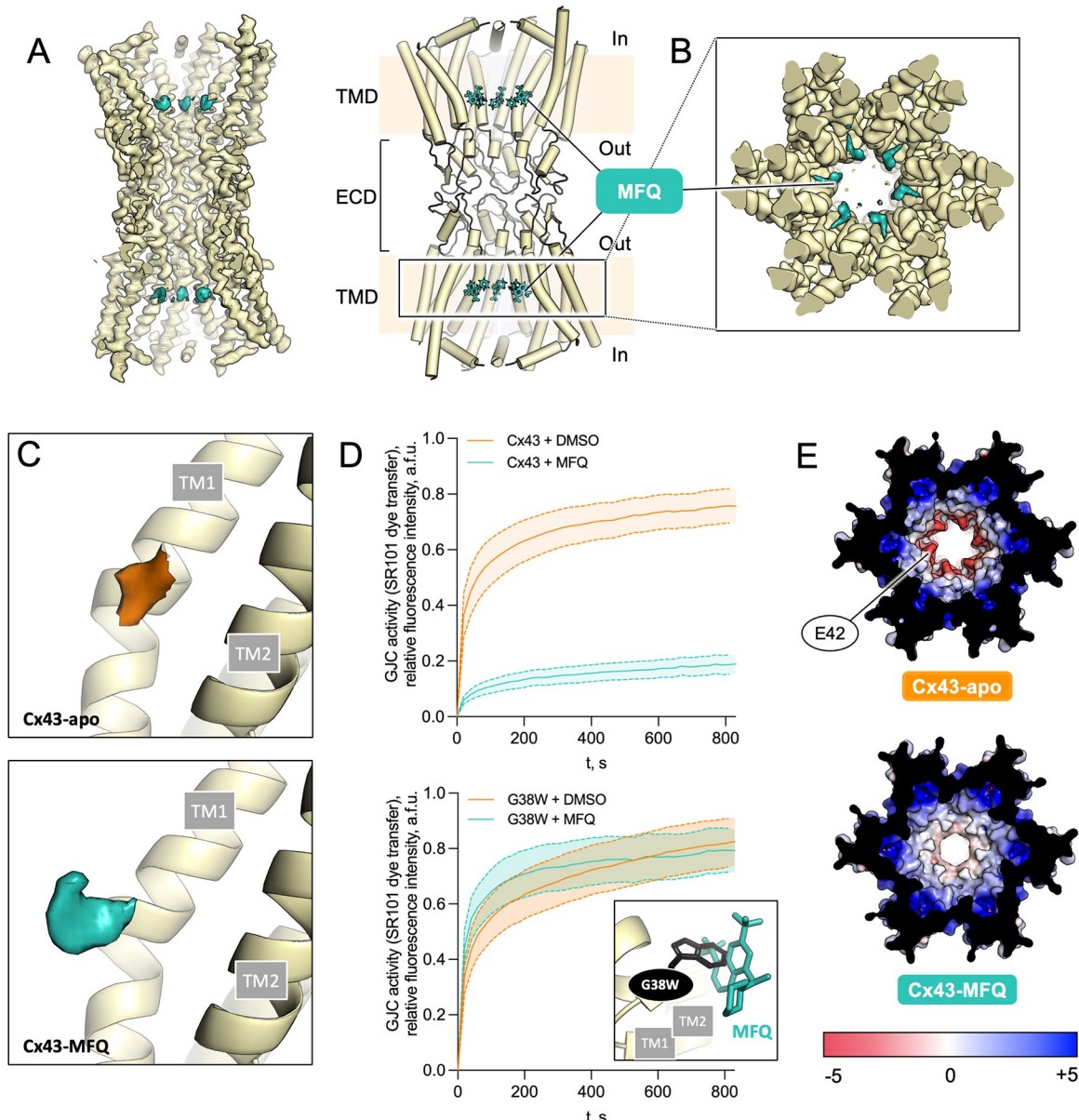

**Fig 5. Functional and structural analysis of MFQ effect on Cx43 GJC.** (**A**) A view of the cryo-EM map and model of Cx43 GJC in complex with MFQ. (**B**) Cryo-EM map of Cx43-MFQ, viewed from the cytosolic side; a clipped view, excluding the NTH density is shown. (**C**) Comparison of site M density of Cx43-apo GJC and Cx43-MFQ GJC (contoured at 3.3σ). (**D**) GJC permeability of WT Cx43 (top) and G38W (bottom) with or without MFQ treatment (n = 15). All data in **D** are represented as mean ± SEM. *Inset*: A G38W mutation sterically hinders MFQ binding, with the side chain of W38 clashing with the ligand. (**E**) Comparison of electrostatic surface potential of Cx43-apo (PDB ID: 7Z1T) and Cx43-MFQ GJCs.

To functionally validate our structural observations in Cx43 we analyzed GJC sensitivity of the G38W mutant to MFQ (**Fig 5D**, *inset*). This mutation is equivalent to V37W in Cx32, which is insensitive to MFQ-mediated inhibition of HC and GJC permeability. The G38W mutant retained the basic function as a GJC, but lost sensitivity to MFQ (**Fig 5D**). In the gap-FRAP assays the ability of MFQ to block the channel function was lost, confirming that site M is the site of inhibitor action not only in Cx32, but also in Cx43.

## Discussion

The structures of Cx32 GJC in complex MFQ and 2APB, and Cx43 GJC in complex with MFQ, along with the high-resolution structures of Cx36 bound to the MFQ analogues [39] shine a light on the molecular mechanisms of connexin channel inhibition. The two drug binding sites (sites A and M) correspond to the previously described lipid binding sites that may bind either sterols or the acyl chains of phospholipid molecules [24,41,43,44]. While binding of phospholipids to site A may stabilize the NTH in a closed conformation [43,44], the functional properties of the site M have not been investigated prior to this study [41,43,44,46]. Our results suggest that drug binding to site A may affect NTH-mediated connexin pore gating (**Fig 6**). Our functional studies as well as previous work on the effect of 2APB on Cx32 HC and GJC permeation [37,47] show that 2APB inhibits Cx32 channels, whereby we observe that its effect is mediated by the observed NTH rearrangements. Thus, the obtained Cx32 GJC reconstruction in complex with 2APB likely represents a closed conformation of the channel.

In contrast to the site A, small molecule binding to site M does not lead to dramatic changes in the pore radius. The pore dimensions of the MFQ-occupied Cx32 or Cx43 leave a substantial opening within the pore region. However, binding of a drug to the six binding sites per connexon within the GJC dramatically affects the electrostatic potential of the pore, introducing two hydrophobic barriers into the solute translocation pathway. A link between pore electrostatics and permeability has been observed in Cx31.3 HC, where introduction of R15G point mutation altered the surface charge at the cytoplasmic pore opening from positive to neutral, without any associated pore radius changes, resulting in increased permeability to ATP [48]. The site M of Cx32 is positioned close to the residue E41 (residues E42 in Cx43, E43 in Cx36; S20 Fig in S1 File), a highly conserved residue among connexins implicated in voltage gating [45,49,50]. The equivalent glutamate residues in Cx26 and Cx46 have been proposed to be a part of the channel's slow gate; the E42Q mutations in Cx26 and Cx46 were shown to abrogate the HC currents [45,49]. It is likely that the changes in electrostatic potential adjacent to the site M / E41 region will dramatically affect channel permeability.

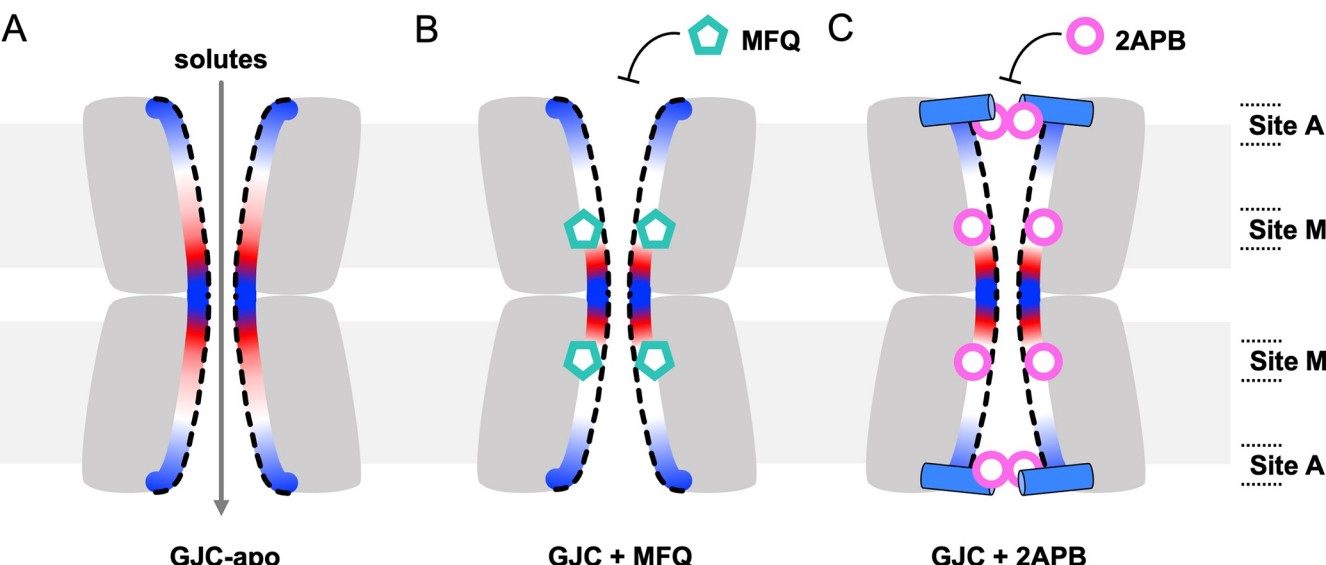

**Fig 6. Mechanism of connexin channel inhibition via sites A and M.** (**A**) In the absence of the inhibitors, the connexin GJCs allow free passage of solutes (<1.5 kDa). (**B**) Binding of MFQ to site M introduces and electrostatic barrier in the pore, reducing GJCs permeability. (**C**) 2APB binds to site M and to site A, causing NTH rearrangement and constriction of the pore entrance.

Although both MFQ and 2APB are relatively weak inhibitors for Cx32 and Cx43, our *in vitro* results are close to the previously reported $IC_{50}$ values (25 μM for MFQ Cx32 GJC coupling inhibition in RIN cells [40] and 47 μM for 2APB inhibition of Cx32 HCs [37]). Nevertheless, the identification of the binding sites of these inhibitors in Cx32 and Cx43 GJCs delivers the first insights crucial for the future development of potent and highly specific chemical agents that act on connexin channels. The connexin inhibitors may be particularly favorable in treatment of connexin HC-related diseases and related pathologies (such as healing of wounds [19], arrhythmia [51], inflammation [52]), which may be normally caused by increased HC function [10,16]. In a growing number of described cases, small molecules that selectively block connexin HCs, but not GJCs, provide the means to protect tissues under pathological conditions. This includes temporal lobe epilepsy [53], skeletal muscle atrophy [54] and spinal cord injury [55]. Moreover, specific drugs capable of uncoupling the GJC-connected cells may be highly desirable particularly in non-metastasized primary tumors [56–58]. Our structures provide a path to a strategy for targeting the connexin channels using structure-based drug design approaches. The site A, which maps to a highly dynamic gating region of the connexin channels, may offer advantages as a binding site more accessible from the cytosol, with the caveat of relatively poor definition of this site in the known connexin channels due to high degree of observed mobility in the NTH. In contrast, the site M is located in a rigid and structurally conserved pocket in each of the connexin channels. The site has very favorable features for designing specific compounds with high selectivity for a connexin of interest. The novel mode of drug binding and inhibition via site M observed in Cx32, Cx43 and Cx36 may offer additional advantages in designing new compounds with unique properties, such as cooperative binding or selective channel permeability.

## Supporting information

**S1 File. Materials and Methods, S1 to S20 Figs, S1 Table and supplementary references (1–20).**
(DOCX)

**S2 File. Mass spectrometry / cell surface biotinylation report (PXD056242), highlighting the expression of Panx1 and Cx32 in HEK293 cells (control cells and Cx32 plasmid-transfected).**
(XLSX)

**S3 File. Source data.**
(XLSX)

## Acknowledgments

We would like to thank M. Peterek (ScopeM, ETH Zurich) and E. Poghosyan (EM Facility, Paul Scherrer Institute) for expert support in cryo-EM data collection, and P. Afanasyev (Cryo-EM Knowledge Hub, ETH Zurich) for data processing advice. We thank S. Bliven and M. Caubet-Serrabou for support in high-performance computing. We thank E. Lazzarin (Medical University of Vienna) for advice on performing APBS calculations.

## Author Contributions

**Conceptualization:** Volodymyr M. Korkhov.

**Data curation:** Volodymyr M. Korkhov.

**Funding acquisition:** Volodymyr M. Korkhov.

**Investigation:** Pia Lavriha, Yufei Han, Xinyue Ding, Dina Schuster, Chao Qi, Anand Vaithia, Volodymyr M. Korkhov.

**Methodology:** Pia Lavriha, Yufei Han, Xinyue Ding, Dina Schuster, Chao Qi, Anand Vaithia, Paola Picotti, Volodymyr M. Korkhov.

**Project administration:** Volodymyr M. Korkhov.

**Supervision:** Volodymyr M. Korkhov.

**Visualization:** Pia Lavriha, Yufei Han, Volodymyr M. Korkhov.

**Writing – original draft:** Pia Lavriha, Yufei Han, Volodymyr M. Korkhov.

**Writing – review & editing:** Pia Lavriha, Yufei Han, Xinyue Ding, Chao Qi, Anand Vaithia, Volodymyr M. Korkhov.

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
