## [Decision Letter · Decision Letter 0]

5 Aug 2024

PONE-D-24-28479Mechanism of connexin channel inhibition by mefloquine and 2-aminoethoxydiphenyl boratePLOS ONE

Dear Dr. % Korkhov,

Thank you for submitting your manuscript to PLOS ONE. After careful consideration, we feel that it has merit but does not fully meet PLOS ONE’s publication criteria as it currently stands. Therefore, we invite you to submit a revised version of the manuscript that addresses the points raised during the review process.

We look forward to receiving your revised manuscript.

Kind regards,

Eliseo A Eugenin, Ph.D.

Academic Editor

PLOS ONE

Journal Requirements:

"The work was supported by the Swiss National Science Foundation (grant 184915) to V.M.K."

4. Please note that funding information should not appear in the Acknowledgments section or other areas of your manuscript. We will only publish funding information present in the Funding Statement section of the online submission form. Please remove any funding-related text from the manuscript.

5. Please note that in order to use the direct billing option the corresponding author must be affiliated with the chosen institute. Please either amend your manuscript to change the affiliation or corresponding author, or email us at plosone@plos.org with a request to remove this option.

6. We note that your Data Availability Statement is currently as follows: 

"The cryo-EM density maps and model coordinates have been deposited to the Electron Microscopy Data Bank and Protein Data Bank with the following accession numbers: PDB ID 8QJF, EMD-18446; PDB ID 8QK6, EMD-18457; PDB-ID 8QJH, EMD-ID 18447; PDB-ID 8QKI, EMD-18463; PDB-ID 8QKO, EMD-18468. All data required to evaluate the conclusions in this paper are present in the paper and in the Supplementary Materials"

7. PLOS ONE now requires that authors provide the original uncropped and unadjusted images underlying all blot or gel results reported in a submission’s figures or Supporting Information files. This policy and the journal’s other requirements for blot/gel reporting and figure preparation are described in detail at https://journals.plos.org/plosone/s/figures#loc-blot-and-gel-reporting-requirements and https://journals.plos.org/plosone/s/figures#loc-preparing-figures-from-image-files. When you submit your revised manuscript, please ensure that your figures adhere fully to these guidelines and provide the original underlying images for all blot or gel data reported in your submission. See the following link for instructions on providing the original image data: https://journals.plos.org/plosone/s/figures#loc-original-images-for-blots-and-gels.   

8. We notice that your supplementary figures/tables are included in the manuscript file. Please remove them and upload them with the file type 'Supporting Information'. Please ensure that each Supporting Information file has a legend listed in the manuscript after the references list.

**Additional Editor Comments:**

Dear Dr. Korkhov:

Thank you for submitting your manuscript to PLOSone. Your manuscript was evaluated for two experts in the field, with significant issues about data presentation and interpretation that requires your attention.

Best regards

Eliseo Eugenin

Reviewers' comments:

Reviewer's Responses to Questions

**Comments to the Author**

1. Is the manuscript technically sound, and do the data support the conclusions?

Reviewer #1: Yes

Reviewer #2: Yes

2. Has the statistical analysis been performed appropriately and rigorously? 

Reviewer #1: I Don't Know

Reviewer #2: Yes

3. Have the authors made all data underlying the findings in their manuscript fully available?

Reviewer #1: Yes

Reviewer #2: Yes

4. Is the manuscript presented in an intelligible fashion and written in standard English?

Reviewer #1: Yes

Reviewer #2: Yes

5. Review Comments to the Author

Reviewer #1: From my recollection gap junctional communication is always necessary for normal tissue function whereas as the hemichannel activity is already shown by numerous groups it could be rather deleterious. In addition, there are known small molecules that do not block gap junction channels and block Cx hemichannels providing a great tissue protection in pathological conditions. These issues should be clarified in the introduction and discussion section as well. Otherwise, the manuscript data is very interesting.

Please see the following references. doi.org/10.1038/s41467-019-14063-8; doi.org/10.1073/pnas.2213162119; doi: 10.3389/fncel.2023.1163436.

During inflammation there is reduction in gap junctional communication and increase in hemichannel activity and hemichannel blockers have been shown to prevent or reduced cell death or tissue dysfunction. Clearly, what is needed to test the role of reduced gap junctional communication are molecules that increase the gap junction channel activity without affecting hemichannels.

Reviewer #2: In this study, Lavriha and colleagues set out to explore the molecular basis of inhibition of Cx32-based channels by MFQ and 2APB. They employed a range of techniques, including dye uptake assays, FRAP analysis, Cryo-EM samples preparation and analysis, immunohistochemistry, tryptophan fluorescence quenching-based binding assays and molecular modeling. Additionally, they expressed and analyzed different mutant constructs for Cx32. Their findings revealed that 2APB binds to "site A," near the N-terminal gating helix of Cx32 GJC, restricting access to the channel pore. In contrast, MFQ binds to a different "site M," which is deeply embedded within the pore. Overall, most of the experiments were well-executed, and the chosen methodologies aligned with the primary research questions.

Concerns:

1. Why did the authors choose 2-APB as an inhibitor of connexin-based channels? Its use is less common compared to other well-known hemichannel and GJC blockers (only 18 results in PubMed using the keywords "2-APB" and "connexin"). In fact, 2-APB is better known as an inhibitor of both IP₃ receptors and TRP channels.

2. Line 65. The authors should cite: PMID: 32612499 and PMID: 33769363

3. Why did the authors choose SR101 instead of other commonly used dyes to assess hemichannel activity (e.g., ethidium, YOPRO, DAPI, etc.)? What evidence supports the ability of this dye to permeate Cx32-based channels?

4. How do the authors rule out the contribution of Panx1 hemichannels to SR101 uptake? Previous studies have shown that HEK293 cells endogenously express Panx1 (PMID: 17036048)

5. Fig. 1. The authors should show representative images for SR101 uptake and FRAP analysis.

6. Does the MFQ or 2-APB absorb at or near the excitation or emission wavelengths of tryptophan at the concentrations required for the titration study? If so, is the inner filter effect significant?

6. PLOS authors have the option to publish the peer review history of their article (what does this mean?). If published, this will include your full peer review and any attached files.

Reviewer #1: No

Reviewer #2: No

---

## [Author Response · Author response to Decision Letter 0]

14 Nov 2024

RESPONSE: We adjusted the style to match the PLOS ONE style. In case any further changes are needed, we will be happy to implement them.

RESPONSE: We provide the correct grant numbers.

"The work was supported by the Swiss National Science Foundation (grant 184915) to V.M.K."

RESPONSE: The cover letter has been updated with the required statement – the funders had no role in study design, data colelciton and analysis, decision to publish, or preparation of the manuscript.

4. Please note that funding information should not appear in the Acknowledgments section or other areas of your manuscript. We will only publish funding information present in the Funding Statement section of the online submission form. Please remove any funding-related text from the manuscript.

RESPONSE: We remored the funding information from the Acknowledgements in the revised manuscript.

5. Please note that in order to use the direct billing option the corresponding author must be affiliated with the chosen institute. Please either amend your manuscript to change the affiliation or corresponding author, or email us at plosone@plos.org with a request to remove this option.

RESPONSE: As a corresponding author, I am affiliated with ETH Zurich (as a professor) and with the Paul Scherrer Institute (as a group leader) – this is a dual appointment. Direct billing can probably be applied with my ETH Zurich affiliation, but I am happy to modify this if necessary.

6. We note that your Data Availability Statement is currently as follows: 

"The cryo-EM density maps and model coordinates have been deposited to the Electron Microscopy Data Bank and Protein Data Bank with the following accession numbers: PDB ID 8QJF, EMD-18446; PDB ID 8QK6, EMD-18457; PDB-ID 8QJH, EMD-ID 18447; PDB-ID 8QKI, EMD-18463; PDB-ID 8QKO, EMD-18468. All data required to evaluate the conclusions in this paper are present in the paper and in the Supplementary Materials"

RESPONSE: We updated the Data Availability Statement with the Mass Spectrometry dataset deposition in the PRIDE database. We also provide all data in the manuscript as File S3 (Source Data). The are no restrictions on the data, and no ethical issues.

7. PLOS ONE now requires that authors provide the original uncropped and unadjusted images underlying all blot or gel results reported in a submission’s figures or Supporting Information files. This policy and the journal’s other requirements for blot/gel reporting and figure preparation are described in detail at https://journals.plos.org/plosone/s/figures#loc-blot-and-gel-reporting-requirements and https://journals.plos.org/plosone/s/figures#loc-preparing-figures-from-image-files. When you submit your revised manuscript, please ensure that your figures adhere fully to these guidelines and provide the original underlying images for all blot or gel data reported in your submission. See the following link for instructions on providing the original image data: https://journals.plos.org/plosone/s/figures#loc-original-images-for-blots-and-gels. 

RESPONSE: We have now included the File S3 (Source Data) with all gel images. The cover letter has also been updated with a corresponding statement.

8. We notice that your supplementary figures/tables are included in the manuscript file. Please remove them and upload them with the file type 'Supporting Information'. Please ensure that each Supporting Information file has a legend listed in the manuscript after the references list.

RESPONSE: The File S1 now contains the supplementary figures / tables. 

Response to reviewer’s comments

5. Review Comments to the Author

Reviewer #1: From my recollection gap junctional communication is always necessary for normal tissue function whereas as the hemichannel activity is already shown by numerous groups it could be rather deleterious. In addition, there are known small molecules that do not block gap junction channels and block Cx hemichannels providing a great tissue protection in pathological conditions. These issues should be clarified in the introduction and discussion section as well. Otherwise, the manuscript data is very interesting.

Please see the following references. doi.org/10.1038/s41467-019-14063-8; doi.org/10.1073/pnas.2213162119; doi: 10.3389/fncel.2023.1163436.

During inflammation there is reduction in gap junctional communication and increase in hemichannel activity and hemichannel blockers have been shown to prevent or reduced cell death or tissue dysfunction. Clearly, what is needed to test the role of reduced gap junctional communication are molecules that increase the gap junction channel activity without affecting hemichannels.

RESPONSE: We are grateful to the reviewer for pointing this out and we have included the references (Cisterna et al., 2020, Guo et al., 2022, Toro et al., 2023) in the introduction. We integrated these reference as follows (page 6, line 265):

“In a growing number of described cases, small molecules that selectively block connexin HCs, but not GJCs, provide the means to protect tissues under pathological conditions. This includes temporal lobe epilepsy [53], skeletal muscle atrophy [54] and spinal cord injury [55].”

Reviewer #2: In this study, Lavriha and colleagues set out to explore the molecular basis of inhibition of Cx32-based channels by MFQ and 2APB. They employed a range of techniques, including dye uptake assays, FRAP analysis, Cryo-EM samples preparation and analysis, immunohistochemistry, tryptophan fluorescence quenching-based binding assays and molecular modeling. Additionally, they expressed and analyzed different mutant constructs for Cx32. Their findings revealed that 2APB binds to "site A," near the N-terminal gating helix of Cx32 GJC, restricting access to the channel pore. In contrast, MFQ binds to a different "site M," which is deeply embedded within the pore. Overall, most of the experiments were well-executed, and the chosen methodologies aligned with the primary research questions.

Concerns:

1. Why did the authors choose 2-APB as an inhibitor of connexin-based channels? Its use is less common compared to other well-known hemichannel and GJC blockers (only 18 results in PubMed using the keywords "2-APB" and "connexin"). In fact, 2-APB is better known as an inhibitor of both IP₃ receptors and TRP channels.

RESPONSE: We were aware of 2-APB as a less well studied and rather poorly defined connexin inhibitor. We were encouraged by the published data that suggested that 2-APB inhibits Cx32 (Tao and Harris 2007, PMID: 17095584), and therefore we decided to investigate this drug in our structural studies. Nevertheless, the reviewer’s point is certainly valid – 2-APB is known to act also on other classes of ion channels. Once we saw that Cx32 reacts so dramatically to 2-APB, with a substantial rearrangement of the N-terminus, we were convinced that the choice of 2-APB for our study was clearly favourable.

2. Line 65. The authors should cite: PMID: 32612499 and PMID: 33769363

RESPONSE: We have included the references PMID 32612499 (Droguerre at al., Front Neurosci, 2020) and PMID 33769363 (Letellier et al., Pain, 2021), as suggested by the reviewer (refs 33 and 34, line 72).

3. Why did the authors choose SR101 instead of other commonly used dyes to assess hemichannel activity (e.g., ethidium, YOPRO, DAPI, etc.)? What evidence supports the ability of this dye to permeate Cx32-based channels?

RESPONSE: We found that SR101 was previously found to be used for Cx32 by Tachikawa et al., 2020 (PMID: 31837976). The spectral properties of SR101 matched well our protein expression constructs, as we routinely use either CFP, YFP as expression markers – either as IRES-based expression markers, or as fusion proteins. Our own tests showed that SR101 is an excellent dye for both HC and GJC assays, so we decided to carry on using this solute in our assays. In our own first publication on Cx32, Qi et al., Sci Adv, 2023, we extensively characterized Cx32 using SR101.

4. How do the authors rule out the contribution of Panx1 hemichannels to SR101 uptake? Previous studies have shown that HEK293 cells endogenously express Panx1 (PMID: 17036048)

RESPONSE: While we can not completely rule out Panx1 hemichannel contribution, we did not explicitly attempt to inhibit Panx1. It is of course possible that the low level background uptake of SR101 is mediated by Panx1 to some extent (and contributions from other membrane proteins present at the cell surface can not be excluded). Two lines of evidence argue against the observed effects due to Panx1 activity: (i) The non-transfected or mock-transfected cells do not show SR101 uptake on the same level as Cx32-transfected cells, (ii) A striking effect of the point mutation in Cx32 (or in Cx43) abolishing the effect of mefloquine on dye uptake / transfer indicates that the effect is specific to our transfected construct. We could clearly establish a strong signal in transfected cells above the background uptake in cells not expressing the exogenous Cx32 - these negative controls gave us the confidence in our assay system.

Most importantly, in the revised manuscript we have now included an additional dataset, analysing the Panx1 expression in our HEK293 cells using mass spectrometry-based proteomics analysis (File S2, detailed in “Materials and Methods”). Our analysis shows that while we can detect low levels of Panx1 in HEK293 cells, the Panx1 expression pattern does not change upon overexpression of Cx32. This confirms that the signal observed in our assays reports the activity of the transfected Cx32 construct, not the endogenous Panx1. A description of this important result is included in the revised text (line 101):

“HEK293 cells are known to express pannexin-1 (Panx1) [42], which could potentially influence the observed dye uptake. To determine whether Panx1 expression is modified by over expression of Cx32, we performed mass spectrometry-based cell surface biotinylation assays. The results confirmed that although Panx1 is indeed expressed in our cells, transfection with the Cx32 expression plasmid does not increase Panx1 expression. The relative expression levels of Cx32 and Panx1 in the control cells were (in arbitrary units): undetected and 41429.81 ± 20171.99, respectively. In Cx32- transfected cells, Cx32 and Panx1 levels were 674598.69 ± 58360.1 and 30106.35 ± 9649.86, respectively (S2 File). Thus, we interpret the observed signals in the HC assays as a consequence of Cx32 expression at the plasma membrane.”

5. Fig. 1. The authors should show representative images for SR101 uptake and FRAP analysis.

RESPONSE: As suggested by the reviewer, we have now included representative supplementary figures for both SR101 uptake and transfer (S1, S2 and S19 Figures in File S1 in the revised manuscript).

6. Does the MFQ or 2-APB absorb at or near the excitation or emission wavelengths of tryptophan at the concentrations required for the titration study? If so, is the inner filter effect significant?

RESPONSE: This was an important question by the reviewer, and we tested this: neither MFQ nor 2-APB absorb at or near the excitation or emission wavelengths of Trp (and consequently there is also no fluorescent signal upon excitation at 295 nm). The figure R1 below shows the ligand spectra, which we include here for the purpose of the response to reviewer comments. The inner filter effect can thus be excluded. In the interest of saving space we prefer to omit this from the revised manuscript.

Figure R1 - included in the response file submitted with the revised manuscript.

---

## [Decision Letter · Decision Letter 1]

27 Nov 2024

Mechanism of connexin channel inhibition by mefloquine and 2-aminoethoxydiphenyl borate

PONE-D-24-28479R1

Dear Dr. Korkhov,

We’re pleased to inform you that your manuscript has been judged scientifically suitable for publication and will be formally accepted for publication once it meets all outstanding technical requirements.

Kind regards,

Eliseo A Eugenin, Ph.D.

Academic Editor

PLOS ONE

Additional Editor Comments (optional):

Dear Dr. Korkhov:

Thank you for submit your manuscript to PLOSone. Thank you for answer the reviewers

Best Regards

Eliseo Eugenin

Reviewers' comments:

Reviewer's Responses to Questions

**Comments to the Author**

1. If the authors have adequately addressed your comments raised in a previous round of review and you feel that this manuscript is now acceptable for publication, you may indicate that here to bypass the “Comments to the Author” section, enter your conflict of interest statement in the “Confidential to Editor” section, and submit your "Accept" recommendation.

Reviewer #2: All comments have been addressed

2. Is the manuscript technically sound, and do the data support the conclusions?

Reviewer #2: Yes

3. Has the statistical analysis been performed appropriately and rigorously? 

Reviewer #2: Yes

4. Have the authors made all data underlying the findings in their manuscript fully available?

Reviewer #2: Yes

5. Is the manuscript presented in an intelligible fashion and written in standard English?

Reviewer #2: Yes

6. Review Comments to the Author

Reviewer #2: The authors have thoroughly addressed all of my major concerns by conducting additional data analyses and enhancing the clarity of the text.

7. PLOS authors have the option to publish the peer review history of their article (what does this mean?). If published, this will include your full peer review and any attached files.

Reviewer #2: No

---

## [Editor Report · Acceptance letter]

6 Dec 2024

PONE-D-24-28479R1 

PLOS ONE

Dear Dr. Korkhov, 

I'm pleased to inform you that your manuscript has been deemed suitable for publication in PLOS ONE. Congratulations! Your manuscript is now being handed over to our production team.

Kind regards, 

on behalf of

Dr. Eliseo A Eugenin 

Academic Editor

PLOS ONE